# Linear regression reporting practices for health researchers, a cross-sectional meta-research study

**Lee Jones**[1,2,3*], **Adrian Barnett**[2], **Dimitrios Vagenas**[1]

**1** Research Methods Group, Faculty of Health, School of Public Health and Social Work, Queensland University of Technology, Kelvin Grove, Queensland, Australia, **2** AusHSI, Centre for Healthcare Transformation, Faculty of Health, School of Public Health and Social Work, Queensland University of Technology, Kelvin Grove, Queensland, Australia, **3** Statistics Unit, QIMR Berghofer Medical Research Institute, Herston, Queensland, Australia

* lee.jones@qut.edu.au

## Abstract

### Background

Decisions about health care, such as the effectiveness of new treatments for disease, are regularly made based on evidence from published work. However, poor reporting of statistical methods and results is endemic across health research and risks ineffective or harmful treatments being used in clinical practice. Statistical modelling choices often greatly influence the results. Authors do not always provide enough information to evaluate and repeat their methods, making interpreting results difficult. Our research is designed to understand current reporting practices and inform efforts to educate researchers.

### Methods

Reporting practices for linear regression were assessed in 95 randomly sampled published papers in the health field from *PLOS ONE* in 2019, which were randomly allocated to statisticians for post-publication review. The prevalence of reporting practices is described using frequencies, percentages, and Wilson 95% confidence intervals.

### Results

While 92% of authors reported p-values and 81% reported regression coefficients, only 58% of papers reported a measure of uncertainty, such as confidence intervals or standard errors. Sixty-nine percent of authors did not discuss the scientific importance of estimates, and only 23% directly interpreted the size of coefficients.

### Conclusion

Our results indicate that statistical methods and results were often poorly reported without sufficient detail to reproduce them. To improve statistical quality and direct health funding to effective treatments, we recommend that statisticians be involved in the research cycle, from study design to post-peer review. The research environment is an

**Data availability statement:** The raw data and a reproducible R Quarto file used to produce this paper, including all tables and figures have been

stored in a GitHub repository and can be accessed at https://github.com/Lee-V-Jones/Linear_Regression_Reporting_Practices.

**Funding:** There was no cost associated with this research except for attending conferences. These costs were covered by the primary author's PhD allocation from the health faculty, Queensland University of Technology, and scholarships. The Statistical Society of Australia (SSA) and the Association for Interdisciplinary Meta-research & Open Science (AIMOS) supported the primary author with travel grants to attend their respective conferences. The funders had no role in study design, data collection and analysis, decision to publish, or preparation of the manuscript.

**Competing interests:** The authors have declared that no competing interests exist.

ecosystem, and future interventions addressing poor statistical quality should consider the interactions between the individuals, organisations and policy environments. Practical recommendations include journals producing templates with standardised reporting and using interactive checklists to improve reporting practices. Investments in research maintenance and quality control are required to assess and implement these recommendations to improve the quality of health research.

## Introduction

Health systems are inherently complex, requiring decisions that balance scientific evidence with practical considerations such as feasibility, equity, and stakeholder perspectives [1]. Central to this process is the role of statistics, which aid in the interpretation of study results to provide the quantitative evidence necessary for informed decision-making about the effects of diseases and treatments. When researchers analyse data, they face many decisions, including which variables to explore, what statistical test to perform, and whether data should be excluded or transformed [2]. It has been increasingly recognised that the transparency of the decisions made through this process plays an essential role in interpreting results [3].

Evidence suggests that poor statistical quality amongst researchers is endemic, with an estimated 85% of medical research avoidably wasted through poor study design, analysis, reporting quality and the low frequency of publication of non significant results [4]. This shocking figure can be attributed to several sources, including 50% of health research not being published [5]; when reported, studies are often poorly designed, inappropriately analysed, and selectively reported, with benefits often exaggerated [6]. While there is a discussion of these issues, such as the publish or perish culture within universities [7] and questionable research practices [8], it is widely acknowledged that lack of statistical training contributes to all aspects of poor reporting [9].

At the centre of the research waste problem is the quality of statistical reporting and the rising importance of p-values. The widespread misuse and misunderstanding of p-values have been reported for decades [10,11], with many researchers mindlessly applying significance rules without understanding the size or importance of the studied effect [12]. King et al. [13] suggest that problems with the selection and interpretation of statistical methods are driven by researchers' reliance on statistical rules of thumb and justification of traditional methods that are popular in the field, even if they are inappropriate. Stark and Saltelli [14] suggest many researchers are guilty of "cargo cult" thinking and go through the process of fitting models, calculating p-values and invoking statistical terms with little understanding of the methods involved.

Reporting guidelines have been created to help address poor reporting and increase transparency and reproducibility in health research. Reporting quality is crucial for assessing the risk of bias. Insufficient information reduces the ability to evaluate bias, undermining the credibility and reliability of the study's findings and limiting their value for guiding clinical practice or informing policy decisions [15]. While many research guidelines exist, very few provide detailed advice on reporting and interpreting data analysis [16]. Examples of statistical guidelines for authors include the Statistical Analyses and Methods in the Published Literature (SAMPL) [17], Strengthening Analytical Thinking for Observational Studies (STRATOS) [18] and Transparent reporting of a multivariable prediction model for individual prognosis or diagnosis (TRIPOD) [19]. The SAMPL was created by Lang et al. [17]

and includes reporting guidelines for common statistical methods, including linear regression. Many authors have recommended the SAMPL guidelines [3,16], but only a small number of studies actually cite SAMPL for either individual use [20,21] or for reporting of quality in reviews [22,23], suggesting they are not widely used, highlighting the need to promote statistical guidelines to increase awareness and use among health professionals [24].

There have been robust conversations within the statistical community on how to improve the quality of statistical reporting, much of which focuses on p-values and their interpretation with calls to either remove p-values entirely and focus on confidence and prediction intervals or use alternative methods such as likelihood ratios or Bayes Factors [11]. While there are many commentaries on improving statistical quality, only a few studies directly assess authors' current practice and statistical understanding [25]. This study aims to understand better where statistical reporting can be improved and inform efforts to educate researchers. We selected regression analyses because it is a widely used method that can provide valuable insights when correctly applied. In a related paper, we examined common misconceptions in the assumptions of linear regression [26]. Building on the previous paper, this research highlights the most common issues health researchers face when interpreting regression analyses and makes recommendations for improving practice.

## Materials and methods

This cross-sectional study was designed to understand the prevalence of statistical reporting behaviours for authors using linear regression, including understanding modelling choices and how often statistics were reported, such as coefficients, confidence intervals, and p-values. This paper contains an overview of the methods for this research, for full details see Jones et al. [26].

### Research question

- How do authors using linear regression report their model and results?

### Sample size

The primary aim of this study was not hypothesis testing but to gain a descriptive understanding of current statistical reporting practices in published manuscripts with a focus on regression assumptions. The study was powered with a 5% margin of error to detect a sample proportion of 0.05 (5%) using a two-sided 95% confidence interval, calculated using exact Clopper–Pearson confidence intervals, using PASS [27]. This sample size was also deemed adequate to understand the prevalence of general statistical reporting behaviours. We estimated 40 statisticians were required to rate the 100 papers, with each paper reviewed twice (40 statisticians × 5 papers = 200 reviews), and five papers per reviewer were thought to be reasonable from our experience and initial feedback.

### Study ethics, statistician recruitment, and consent

This study was granted Negligible-Low Risk Ethics from the Queensland University of Technology (QUT) Human Research Ethics Committee with approval number 2000000458. Statisticians were recruited through professional societies in Australia and internationally, such as the Statistical Society of Australia, universities, and other relevant organisations using targeted emails, LinkedIn, and Twitter. The inclusion criterion was previous or current employment as a statistician, data analyst, or data scientist. Study information, including the study protocol, participant information sheet, and the study questions, were available on GitHub

and emailed to participants [28], who were also sent online links to the five *PLOS ONE* papers to be reviewed. Informed written consent was obtained from the statistician by filling out and returning the consent form, which asked if participants would like to be acknowledged in the paper. Study recruitment started in September 2020 and finished in June 2021, with the last participant completing their reviews in September 2021. On average, the median time to completion was four weeks; statisticians were recruited until all 200 reviews were complete. In total, 46 statisticians were recruited, of which five withdrew due to changed circumstances, and one participant had difficulty completing the online form and was replaced.

### Randomisation

One hundred research papers (excluding editorials and other non-research papers) were randomly selected from *PLOS ONE* in 2019. This study was conducted in 2020/21, and 2019 was chosen to gain a current snapshot of reporting practices. While not planned, this research coincided with the full year before the COVID-19 pandemic, avoiding any disruption that may have been caused by changes in the research environment due to the shutdowns [29]. Papers were selected if they had "health" anywhere in the subject area and used the term 'linear regression' in the materials and methods section by using the "searchplos" function within the "rplos" package in R [30]. Papers that met the inclusion criteria were randomly ordered, and the first eligible 100 were selected. To capture the broader use of regression from the population of health researchers, we chose to focus on standard linear regression; papers were excluded if the regression included cluster or random effects, or used alternative methods such as Bayesian, non-parametric, or where the linear regressions were not part of the paper's primary analyses, e.g., related to pre-processing such as calibrating a reference sample.

### Calculating the prevalence of statistical reporting behaviours

Two volunteer statisticians rated each paper, and the primary author, LJ, also independently provided a third statistical rating. The study was initially designed for the prevalence to be calculated using the two ratings with the primary author adjudicating differences; however, due to the length and complexity of papers, it was decided by the authorship team to use all three ratings to improve the accuracy of results. The reliability of ratings from the two independent statisticians was calculated. Then, each set was compared to the final prevalence to assess the impact of the change to the protocol. Disagreements between the three ratings were documented by reading and commenting on the PDF of papers and recording each disagreement. Finally, the results were also cross-checked for consistency; for example, if a paper was identified as having only univariate models (i.e. single explanatory variable), it did not require checks for collinearity. The paper was then checked, and prevalence was updated accordingly.

### Data analysis

The purpose of this study is a descriptive analysis of statistical reporting behaviours, which were described using frequencies, percentages and 95% Wilson confidence intervals to account for percentages close to zero. The reliability of statistical ratings was described using observed agreement and analysed with Gwet's statistics [31]. The assumptions in Gwet's analysis do not require testing but instead, relate to the interpretation and generalisability of results. In this case, papers were randomly sampled and randomly allocated to statisticians; no weighting was applied as variables were either binary or nominal. The STROBE guideline for reporting cross-sectional studies was used [32]. R version 4.4.1 [33] was used for all statistical analyses.

## Linear regression

This section provides some technical background on linear regression, a method widely used in research, for readers who may be unfamiliar with it. It also provides context on what researchers need to report when applying this method in their studies.

Simple linear regression is a statistical method that can be used to understand the relationship between two continuous variables, for example, age and blood pressure. A linear relationship is assumed between the dependent (often notated Y) and model parameters associated with the explanatory variable. The explanatory variables are usually denoted by the X variable, as shown in (Fig 1) and described by (Eq 1). This can be readily expanded to "multiple" regression, which allows for multiple independent variables ($k$ explanatory variables and as many parameters) in the model (Eq 2). This enables us to estimate these model parameters for one variable while taking into account the effect that other explanatory variables can have on this relationship. It also allows the exploration of more complex relationships, such as interactions between explanatory variables. Linear regression can also be used to model categorical X variables. In fact, t-tests, ANOVA and linear regression are special cases of the General Linear Model (GLM), where X variables can be either continuous or categorical.

$$\hat{Y}_i = \hat{\beta}_0 + \hat{\beta}_1 X_i + \hat{\epsilon}_i, \quad i = 1, \dots, N \tag{1}$$

$$\hat{Y}_i = \hat{\beta}_0 + \hat{\beta}_1 X_{1i} + \hat{\beta}_2 X_{2i} + \dots + \hat{\beta}_k X_{ki} + \hat{\epsilon}_i, \quad i = 1, \dots, N \tag{2}$$

(Eq 1) gives the mathematical form of the estimated linear regression. The index "$i$" is for each observation in the data, of which there are $N$ in total. $\hat{\beta}_1$ is the slope; in our example, it represents the average change in blood pressure with a one-unit change in age. The term $\hat{\beta}_0$ is the Y-intercept, which in our example is the blood pressure value when age equals zero. Finally, $\hat{\epsilon}_i$ is the "error" or "residual" term, which is the part of $Y_i$ that cannot be accounted for by the available information, i.e. by $\hat{\beta}_0 + \hat{\beta}_1 X_1$ for each observation.

Good reporting practice involves not only presenting the numbers but also interpreting the results, contextualising their importance, and addressing why they matter. Below (Tables 1 and 2) is an example of the write–up and coefficients table for good reporting of the simulated data presented in Fig 1.

**Regression coefficients and $R^2$.** In a simple linear regression model, the regression coefficient (b) represents the average change in the dependent variable (Y) for every unit increase in the explanatory variable [35]. A common problem interpreting regression coefficients occurs when continuous variables are on a very large or small scale, and it becomes difficult to interpret clinically meaningful change; an easy way to improve interpretation is to scale the variable appropriately. For example, weight in grams can be divided by 1000 and interpreted as the average change in Y with unit change in kilograms. Regression coefficients can also be "standardised" and used to compare variables measured on different scales, and coefficients

**Table 1. Example write–up for blood pressure.**

Blood pressure was found to be associated with age (F(1, 198) = 262.3, p < 0.001), with blood pressure increasing by 1.03 mmHg (95% CI: 0.91, 1.16) per year or 10.3 mmHg as age increases by 10 years (Table 2). Age explained approximately 57% ($R^2$ = 0.57) of the variance in blood pressure. Even small increases in blood pressure can have a significant impact on cardiovascular health, as sustained elevations contribute to an increased risk of adverse outcomes such as heart attack and stroke. For example, in middle age, a 2 mmHg increase in systolic blood pressure is associated with a 10% higher risk of ischemic heart disease and a 7% higher risk of stroke [34].

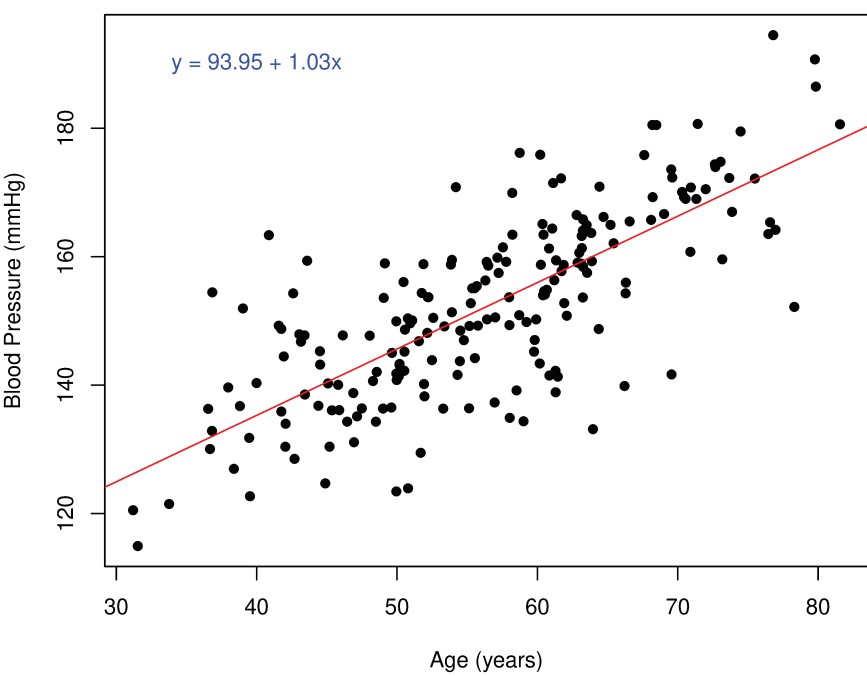

**Fig 1. Example data on age and blood pressure, with the fitted line from a linear regression model.**

**Table 2. Example of a linear regression coefficients table for the univariate relationship between blood pressure and age.**

| | | | | 95% CI | | |
|---|---|---|---|---|---|---|
| Terms | B | SE | t | Lower | Upper | p-value |
| (Intercept) | 93.95 | 3.67 | 25.62 | 86.72 | 101.18 | <0.001 |
| age | 1.03 | 0.06 | 16.20 | 0.91 | 1.16 | <0.001 |

B = Unstandardised coefficient; SE = Standard Error; CI = Confidence Interval

$R^2$ = 0.57; Residual standard error = 9.60; Residual df = 198; No. Obs. = 200

can be interpreted in terms of standard deviations. Suppose the explanatory variable is standardised by subtracting its mean and dividing by its standard deviation. This transformation results in a variable with a mean of 0 and a standard deviation of 1, facilitating comparability across variables. In the univariate case (i.e. a single X), the covariance between standardised Y and standardised X equals the correlation coefficient of the two variables. The square of this correlation coefficient measures the proportion of variance in Y explained by X, given by the R-squared. In this univariate situation, R-squared is equivalent to squaring the correlation [35].

While these relationships between correlation and regression exist, researchers may not appreciate that they become complicated when there is more than one variable in the model and are calculated and interpreted differently. For example, the standardised regression coefficients in a multiple linear regression represent the unique contribution of each independent variable for the prediction of the dependent variable after accounting for the effects of all other variables in the model [35]. $R^2$, known as the coefficient of determination, in this case, represents the proportion of the variance in the dependent variable explained by all the explanatory variables. To account for variance explained by chance (i.e. spurious correlation)

when multiple explanatory variables are in the model, an "adjusted" $R^2$ is used. $R^2$ can be used as an effect size as, in general, a higher $R^2$ value indicates a stronger relationship between the dependent and independent variables. However, it has limitations as it does not provide information on practical significance and considers all variables in the model, rather than specific variables of interest. Therefore, it is recommended that both regression coefficients and $R^2$ are reported.

**P-values, confidence intervals, and scientific importance.**  P-values are the most frequently used measure of statistical evidence across all fields of science and research [36]. Despite the frequency of use, understanding p-values is elusive to most users, with widespread misuse being well-documented since the 1940s [36,37]. When conducting a hypothesis test, a test's significance level (alpha) is chosen to determine the acceptable type I error (falsely rejecting the null hypothesis). The p-value is the probability of obtaining a result at least as extreme as the observed result, assuming the null hypothesis is true. P-values were introduced in the 1920s by Ronald Fisher and were not meant to be a conclusive test but, instead, a way of determining whether a result deserved further investigation [38]. Unfortunately, in practice, researchers often use this continuous measure as a threshold, creating a dichotomy of results declared either statistically significant ($p < 0.05$) or not [39], regardless of practical importance. Despite much work in this area, errors in the logic of p-values remain prolific in the literature [40,41]. To avoid over-interpretation of p-values, it is recommended that the smallest clinical improvement considered consequential to the patient [42] be identified before undertaking the study. In practice, this value may not be known for many exploratory studies, but such studies should still consider practical significance.

When translating research from the lab into clinical practice, researchers should be cautious about making important clinical changes based on the results of one study, as a sample-to-sample variation will likely change the estimated effect [43]. When other scientists replicate the research, the range of coefficients, confidence intervals, and p-values is gained. Usually, researchers don't have access to these replications and must make the best decisions based on the available information [43]. While the width of a confidence interval indicates how much these confidence intervals may bounce around when an experiment is replicated, the p-values fluctuate widely, and they are less useful in understanding whether results will be replicated in future experiments [43]. Therefore, it is recommended that confidence intervals are reported with p-values.

**Data transformation.**  Data transformation is used across the health area when data are skewed or do not fit a normal distribution, which is the distribution assumed for the residuals of linear regression. Data transformation is one tool in the statistical toolbox, and while it is helpful in certain situations, it should be used cautiously. Logarithmic transformations have been used as a cure-all for assumption violations; for a detailed explanation of regression assumptions and outliers, see Jones et al. [26]. When one or both variables have been log-transformed, the interpretation of regression coefficients changes from a unit change to a percent change. Means and 95% confidence intervals of groups can also be back transformed (geometric mean) [44]. When the data fits the underlying transformed distribution, and the residuals of the linear model are normally distributed, the interpretation of results may be improved. However, when there is a lack of fit of the transformed variable, transformations can cause more problems than they fix, as they tend to reduce the variance [45]. Once transformed, interpretation becomes more complex and may distort relationships between variables, and researchers should consider using alternative statistical methods that are more appropriate for their data [45]; for example, a gamma distribution can be used for heavily skewed continuous data.

**Modelling.**   Broadly speaking, there are two approaches to model building: statistical and epidemiological. Statistical approaches include algorithmic methods such as stepwise modelling or regularisation methods that reduce the risk of overfitting [46]. Epidemiological approaches include choosing the final model based on previous literature or known disease pathways, regardless of statistical significance. These approaches may result in different final models, p-values, and parameter estimates [3].

The choice of model for data should be based on the study design and the research question. For example, in a Randomised Controlled Trial (RCT), where participants are successfully randomised into two groups, the only systematic difference between the groups is the study intervention. In practice, while RCTs can be analysed with a simple t-test, there are often adjustments for stratification and other pre-specified variables, all of which should be detailed in the study protocol. In comparison, observational studies are often complex and may have differing purposes depending on the research question. Relationships in observational studies are more difficult to directly measure due to confounding variables, which may distort relationships [47]. If not adequately accounted for, confounding variables may hide the true association between dependent and independent variables, leading to biased estimates and inflation of the variance [48], which will affect subsequent interpretation.

Model selection methods use different approaches to identify the best subset of variables that predict the dependent variable [49]. The most common statistical modelling approach is stepwise selection, which includes backward selection, forward selection, and a combination of both, known as stepwise selection [35]. These approaches iteratively fit models by adding or removing variables based on predefined criteria [49]. However, these methods have been criticised for producing overfitted models that describe the sample well but are less generalisable to the target population [49]. Regularisation methods such as Lasso (Least Absolute Shrinkage and Selection Operator) and Ridge regression are another approach. These methods penalise the model based on complexity by introducing a parameter that allows variable coefficients with minor contributions to be shrunk towards zero [50]. These methods can deal with highly correlated independent variables, with Lasso allowing model selection by shrinking model parameters to absolute zero [50]. All modelling choices should match the study's objective and be pre-planned in a study protocol to allow transparency and avoid p-hacking [18,51].

**Multicollinearity.**   Regression models should be assessed for multicollinearity when multiple independent variables are included. Multicollinearity occurs when high correlations exist between two or more independent variables [35] that explain the same variance in the dependent variable and make it challenging to separate the importance of individual variables. It can lead to unstable coefficients and increased type II errors (incorrectly concluding that the variable is not statistically significant), with the standard errors and confidence intervals becoming inflated [52]. Diagnosis of multicollinearity includes consideration of the Variance Inflation Factor (VIF) and pairwise correlations and examining changes in the standard error of models [53]. VIF measures how much the variance of the estimated parameter is increased due to collinearity, with a rule of thumb of values of 10 indicating a problem [35]. However, Zurr et al. [53] suggest that lower values of three can also indicate problematic collinearity. Treatment for multicollinearity may include using alternate methods such as regularisation methods or dropping one of the variables found to be highly correlated [52]. Deciding which variables should be removed from the model can be done in several ways, including dropping variables with the highest VIF or preferably using clinical understanding to keep the most important predictors in the model.

## Results and discussion

In 2019, 16,318 papers were published in *PLOS ONE*; of these, 1005 (6%) mentioned linear regression in the methods section with health in the subject area. Papers were randomly selected and reviewed for the inclusion criterion until we had 100 papers that used linear regression. Whilst reviewing the paper, the statisticians could exclude papers by indicating no linear regression results reported; this option was provided to reduce the risk of excluding papers with poor reporting. Ten papers were identified as potentially having no linear regression results; the author team reviewed these papers and excluded five. Therefore, 95 papers were included in the final analysis (Fig 2). For papers with at least one result presented but rated by one statistician not to contain linear regression results, the missing review was replaced by the primary authors' results for the reliability analysis.

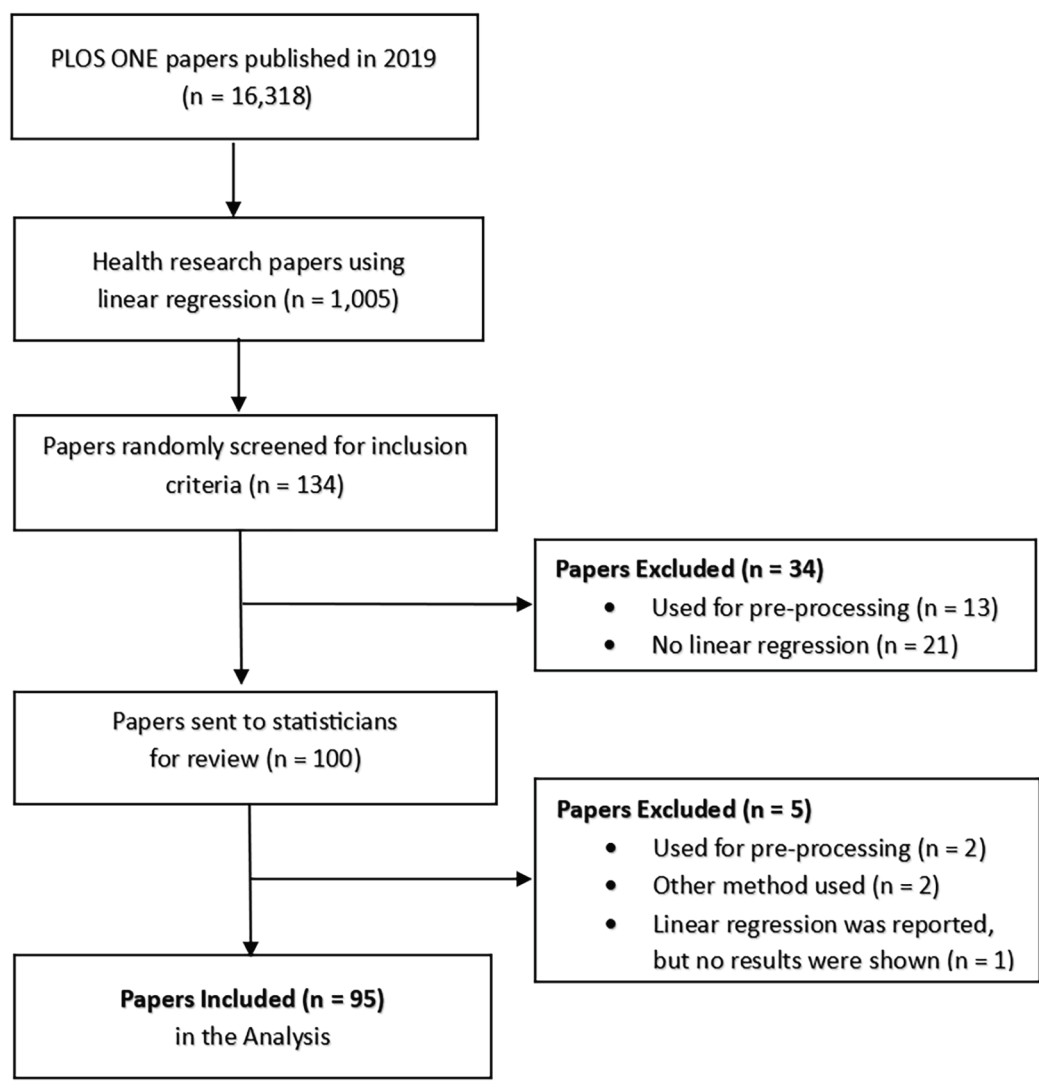

**Fig 2. Flow diagram of the included papers reproduced from Jones et al. [26].**

The majority of the included studies were observational (n = 80, 84%), with 15 experimental studies. Human participants were primarily used in 77% (73) of studies, 13% (12) used animals, 3% (3) used other studies, and the remaining seven studies had either a combination of human, animal, or plants, or were conducted in a lab.

Over half (55%) of statisticians indicated their highest statistical or mathematical education was a PhD, with 28% having a Master's level qualification. Ten percent had honours or bachelor's degrees. One statistician had a diploma, while two others had no formal statistical education. Statisticians were experienced, with 25% having 5–9 years of experience, 30% with 10–19 years, and 23% having 20+ years of experience. For further information on statistician demographics, see Jones et al. [26].

### Prevalence of statistical reporting behaviours

Most authors (92%) used p-values to describe their results, with 17 authors mostly reporting p-values categorically. In comparison, 81% of authors reported regression coefficients, with less than half reporting either confidence intervals (46%) or standard error (20%). The total number of observations was only clear in 47% of papers (Table 3). Thirteen of the 30 studies that transformed their data did not provide any reasoning. Several authors demonstrated poor

**Table 3. Observed prevalence and 95% confidence intervals for reported statistical behaviours, N = 95.**

| Variables | n (%) | 95% CI |
|---|---|---|
| Coefficients | 77 (81%) | 72%, 88% |
| Confidence intervals | 44 (46%) | 37%, 56% |
| Standard error | 19 (20%) | 13%, 29% |
| $R^2$ | 38 (40%) | 31%, 50% |
| F/t statistics | 16 (17%) | 11%, 26% |
| Degrees of freedom | 8 (8%) | 4%, 16% |
| Number of observations in models | 45 (47%) | 38%, 57% |
| Has the direction of the parameter estimates been interpreted? | 63 (66%) | 56%, 75% |
| Has the size of the parameter estimates been interpreted? | 22 (23%) | 16%, 33% |
| Have p-values been reported? | | |
| No | 8 (8%) | 4%, 16% |
| Mostly reported categorically | 17 (18%) | 11%, 27% |
| Mostly reported continuously | 70 (74%) | 64%, 81% |
| Have authors discussed the scientific importance of parameter estimates? | | |
| No | 66 (69%) | 60%, 78% |
| Yes, but only in a generic way | 18 (19%) | 12%, 28% |
| Yes, linked size of effect back to outcome variable | 11 (12%) | 7%, 20% |
| Was collinearity of X variables in models evaluated? | | |
| Not required | 22 (23%) | 16%, 33% |
| No | 60 (63%) | 53%, 72% |
| Yes | 13 (14%) | 8%, 22% |
| Were any continuous variables transformed not including categorisation? | | |
| No | 65 (68%) | 59%, 77% |
| Yes, but did not describe reasoning for transformation | 13 (14%) | 8%, 22% |
| Yes, described reasoning for transformation | 17 (18%) | 11%, 27% |
| Were continuous variables on a very large or small scales in the model scaled appropriately? | | |
| Not required | 50 (53%) | 43%, 62% |
| Unclear | 9 (9%) | 5%, 17% |
| No | 5 (5%) | 2%, 12% |
| Yes | 31 (33%) | 24%, 43% |

N = Number of papers; n (%) = Prevalence; 95% CI = Wilson 95% confidence intervals.

reporting practices, interchangeably using correlation coefficients (r), coefficient of determination ($R^2$), and regression coefficients often without interpretation. Nineteen percent of researchers were unclear about their modelling choice, with 14% using a recognised modelling approach, with less than half of these papers providing sufficient detail to be reproduced (Table 4).

## Agreement of statistical raters

There was a high agreement between the two statistical ratings for reporting behaviours, including coefficients, confidence intervals, test statistics, degrees of freedom and measures of uncertainty (Table 5, for full reporting, see S1 Table). However, lower but moderate agreement (Gwet ≥ 0.4 to 0.59) was observed for questions requiring interpretation by raters, including the size and direction of parameter estimates, p-values, collinearity, and transformation. Relatively poor agreement was observed for some outcomes, including the importance of parameters, variables scaled appropriately, and any process for selecting the variables included in the final model. Reasons for disagreement included differences in considering what was described in both the text and tables. Raters were sometimes split between unclear and not required. Some disagreement between ratings were also due to the authors' methods sections, which were often unclear. In general, the two statistical raters had a higher agreement with the final prevalence score (which took into account the third rating conducted by the primary author) than they did with each other, indicating that while there was variability, the overall prevalence was reflective of raters.

**Table 4. Observed prevalence and 95% confidence intervals for model selection and reporting, N = 95.**

| Variables | n (%) | 95% CI |
|---|---|---|
| Is there any process for selecting the variables included in the final model? | | |
| Unclear | 19 (20%) | 13%, 29% |
| Univariate modelling only (one X variable) | 22 (23%) | 16%, 33% |
| Model based on reference literature or author knowledge | 36 (38%) | 29%, 48% |
| Significant variables from univariate analysis were included in a multivariable model | 3 (3%) | 1%, 9% |
| Used recognised statistical modeling strategy | 13 (14%) | 8%, 22% |
| Other | 2 (2%) | 1%, 7% |
| Which variable selection strategy was used? | | |
| No recognised modelling strategy | 82 (86%) | 78%, 92% |
| Forwards | 2 (2%) | 1%, 7% |
| Backwards | 4 (4%) | 2%, 10% |
| Stepwise | 7 (7%) | 4%, 14% |
| Information criterion | 0 (0%) | 0%, 4% |
| Regularisation methods | 0 (0%) | 0%, 4% |
| Other | 0 (0%) | 0%, 4% |
| Does the paper mention any statistical significance criteria for including variables? | | |
| No recognised modelling strategy | 82 (86%) | 78%, 92% |
| No | 7 (7%) | 4%, 14% |
| Yes | 6 (6%) | 3%, 13% |

N = Number of papers; n (%) = Prevalence; 95% CI = Wilson 95% confidence intervals.

**Table 5. Agreement and reliability of statistical raters for 95 papers.**

| Variable | Rating 1 vs Rating 2 | | | Rating 1 vs Prevalence | | | Rating 2 vs Prevalence | | |
|---|---|---|---|---|---|---|---|---|---|
| | Agree | Gwet | 95% CI | Agree | Gwet | 95% CI | Agree | Gwet | 95% CI |
| Degrees of freedom | 92% | 0.91 | 0.84, 0.98 | 96% | 0.95 | 0.90, 1.00 | 92% | 0.90 | 0.84, 0.97 |
| Variable selection strategy | 92% | 0.91 | 0.85, 0.97 | 94% | 0.93 | 0.88, 0.99 | 89% | 0.89 | 0.82, 0.96 |
| Model significance criteria | 91% | 0.90 | 0.83, 0.97 | 92% | 0.91 | 0.84, 0.97 | 91% | 0.90 | 0.83, 0.96 |
| F/t statistics | 87% | 0.82 | 0.72, 0.93 | 92% | 0.89 | 0.80, 0.97 | 89% | 0.85 | 0.75, 0.95 |
| Standard error | 86% | 0.79 | 0.66, 0.91 | 92% | 0.88 | 0.79, 0.96 | 91% | 0.85 | 0.75, 0.95 |
| Coefficient | 85% | 0.78 | 0.66, 0.90 | 94% | 0.91 | 0.83, 0.98 | 92% | 0.87 | 0.78, 0.96 |
| Confidence intervals | 85% | 0.71 | 0.56, 0.85 | 88% | 0.77 | 0.64, 0.90 | 88% | 0.77 | 0.64, 0.90 |
| R-Squared | 82% | 0.65 | 0.50, 0.81 | 89% | 0.79 | 0.67, 0.92 | 84% | 0.70 | 0.55, 0.84 |
| p-values | 75% | 0.67 | 0.54, 0.79 | 86% | 0.82 | 0.73, 0.92 | 83% | 0.78 | 0.68, 0.89 |
| Direction interpreted | 72% | 0.44 | 0.26, 0.63 | 80% | 0.63 | 0.47, 0.79 | 77% | 0.55 | 0.38, 0.72 |
| variables transformed | 69% | 0.61 | 0.48, 0.75 | 78% | 0.72 | 0.60, 0.83 | 78% | 0.71 | 0.59, 0.83 |
| Size interpreted | 65% | 0.43 | 0.23, 0.62 | 76% | 0.62 | 0.46, 0.78 | 73% | 0.55 | 0.38, 0.73 |
| Collinearity evaluated | 61% | 0.45 | 0.30, 0.59 | 77% | 0.68 | 0.56, 0.80 | 72% | 0.60 | 0.47, 0.74 |
| N in models | 57% | 0.16 | -0.05, 0.37 | 74% | 0.49 | 0.31, 0.67 | 71% | 0.41 | 0.22, 0.60 |
| Importance of parameters | 48% | 0.27 | 0.11, 0.42 | 66% | 0.54 | 0.40, 0.69 | 58% | 0.42 | 0.27, 0.57 |
| Scaled appropriately | 40% | 0.24 | 0.11, 0.38 | 52% | 0.39 | 0.25, 0.53 | 46% | 0.32 | 0.18, 0.45 |
| Process variable selection | 37% | 0.25 | 0.13, 0.36 | 59% | 0.51 | 0.39, 0.63 | 48% | 0.39 | 0.26, 0.51 |

Agree = Observed agreement, Gwet = Gwet agreement coefficient; 95% CI = Gwet 95% confidence intervals.

## Comparison of results to other research

Most authors of the papers included in the current study (81%) reported regression coefficients, however, less than half (46%) of authors used confidence intervals or standard errors (20%) with 58% of papers reporting some measure of uncertainty; 18% of authors mostly reported p-values categorically, ignoring the widely discussed guidelines on p-values from the American Statistical Association [11]. Poor quality of reporting in published work can be seen across all health research fields [36,54]. A Study by Strasak et al. [55] compared two prominent medical journals, *Nature Medicine* and *The New England Journal of Medicine*, in 2004 for common statistical errors. In a subset of 53 papers across the two journals, *The New England Journal of Medicine*, 45% did not report confidence intervals for the main effect, and 19% of articles did not report exact p-values. For *Nature Medicine papers* 95% did not report confidence intervals, and 86% did not report exact p-values. While the study by Strasak et al. [55] occurred before the publication SAMPL guidelines in 2013 [17], Fernandez–Nino and Hernandaz–Montes [56] assessed 108 papers from 2000 to 2017 in three six-year periods, finding an increased volume of reported regressions, with most items on their checklist worsening over time. Our study also showed a high prevalence of poor reporting, indicating that introducing guidelines like SAMPL without enforcement does not resolve poor reporting issues.

The results from our *PLOS ONE* sample were that 23% of studies were univariate, which is a higher use of multivariable modelling than in previously reviewed health literature but in line with the increasing complexity of modelling over time [55]. Real et al. [57] examined the use of multiple regression models in observational studies in Spanish scientific journals between 1970 and 2013. They found only 6.1% of the articles included the term multivariable analysis, with increasing frequency reported from 0.1% in 1980 to 12.3% in 2013.

Although many model selection methods exist, some are more robust than others. Sun et al. [58] outline the danger of using univariate analysis with a statistical significance threshold ($p < 0.05$) as a screening tool for inclusion in multivariable models. The authors provide examples of simulated and real data where this method ignores confounding and

inappropriately excludes important variables, leading to incorrect conclusions about associations. In a review of oncology studies, Mallet et al. [59] found that out of 43 studies, 21 (49%) used univariate analysis as a pre-screening test to select variables for multivariable models. The current research results show a lower proportion of use of the significance threshold as a pre-screen, with 3 out of 64 papers that used multivariable modelling identified that they took this approach. While this sounds like a positive result, it is difficult to interpret as 20% of studies were unclear when describing their modelling process, it was often unclear if models were univariate or multivariable. Statistical sections were often generic and difficult to follow, with poor reasoning, with authors from one paper describing their model selection as the overfitted model, not understanding that fitting a model with a small sample size with many variables is poor practice. Variable inclusion was based on literature or author knowledge in 36 (38%) papers, generally with very little or no explanation. Thirteen (14%) studies used a recognised modelling strategy; of these, only stepwise methods were used, with 6 describing any statistical criteria.

Poor reporting of statistical sections is common in health, with White et al. [60] finding that many papers' content resembled "boilerplate text" cut and pasted from already published work, with often little resemblance to the analyses conducted. Collyer et al. [61] conducted a qualitative study to understand researchers' understanding of linear regression, finding that the interpretation of regression coefficients was described by researchers as iterative and nuanced rather than complete or authoritative statements, which sometimes depended on prior understandings. However, in our study, it was challenging to judge authors understanding as most results were not interpreted, with only 11 (12%) author teams properly linking the size of the effect back to the dependent variable, and another 18 (19%) doing so generically.

Authors rarely report checking for multicollinearity with Vatcheva et al. [62] searching the epidemiological literature in *PubMed* from January 2004 to December 2013 and found that only 1 in 100 regression papers mentioned collinearity or multicollinearity. The authors report that when variables are strongly collinear the normal interpretation of a regression coefficient of a change in Y with a one-unit increase in X while holding the other predictors constant becomes practically impossible. They concluded that although the multicollinearity diagnosis does not solve the problem, it is important to understand the impact on findings and allow greater care to be taken when interpreting the regression coefficients [62]. While our study did not distinguish whether the regressions were conducted for prediction or statistical inference, this is an important distinction because the implications of multicollinearity depend on the analysis objective. In causal/associational models, multicollinearity can distort coefficient estimates and hinder their interpretation, whereas prediction models prioritise overall accuracy and may accommodate correlated predictors if predictive performance remains unaffected. Norstrom et al. [63] reviewed 41 papers from public health and found that only one article was tested for collinearity. Fernandez–Nino and Hernandaz–Montes [56] found that 15% (17/113) of models in *Biomedica* were assessed for collinearity. Our results confirm that proper reporting of collinearity remains poor, with 13 out of 64 (20%) multivariable papers reported checking for collinearity.

## System-wide reform of statistical practices

While blaming individual researchers for poor statistical quality is tempting, our results, which align with previous research [64], indicate systemic issues in understanding and reporting the broader statistical theory and tolerance of bad behaviour [65]. This research aims not to name and shame individual researchers for their reporting practices but to understand the

magnitude of the problem and help guide the culture change required to improve reporting. The broader purpose of this research is not about the individual researchers but rather the practical implication of arriving at the wrong conclusions when bad statistical practices are used. It is about patients and the impact that potentially ineffective treatments might have. Leek et al. [66] suggest that if we think of poor research practices as a disease, we should see the review process as medication with the research quality crisis seen from a primary prevention perspective. As in health prevention, editorial review (medication) is the last step and should not be relied on to fix the problems [66]. Instead, greater investment in prevention strategies such as increasing awareness of issues, increased training and access to qualified methodologists is required to encourage healthy research practices.

Parallels can be observed between poor statistical practices and addictive behaviours, including obesity, where there is a clear relationship between junk food availability in communities [67] (seen as obesogenic environments) with the parallel that some journals can be seen as selling junk (quick publication without adequate review), with institutions rewarding quantity over quality (calorie-rich food deficient in nutritional value). In the same way, telling someone to lose weight won't solve the obesity crisis; just telling people to do better research or to stop abusing p-values will not solve the statistical quality crisis. Like an addictive drug, the reward for poor quality research can create a feedback loop; with more publications required to achieve promotion or funding success, the more shortcuts are taken.

There are complex causes for poor reporting quality observed historically [4] and in the current study. Many opinion pieces have been written on the topic, as well as primary research targeting particular items of statistical reporting, such as p-values and confidence intervals, which has been limited in improving the interpretation of results [36,68]. There are also structural issues with journals focusing on word and table limits rather than good reporting, with poor reporting reinforced by journals requiring short statistical sections rather than comprehensive and transparent reporting. Our study highlighted that even when there are no word limits, statistical sections are often not reported in enough detail to reproduce, suggesting that this may be a learned behaviour. There are no easy solutions, and we recommend a system-wide approach to reform statistical practices. When designing future interventions to tackle poor statistical quality, meta-researchers should incorporate the knowledge from health about behaviour change [69] and complex relationships between individuals and the interplay between interpersonal support systems, the community, the organisation and the policy environment [70]. The social-ecological model proposed by Bronfenbrenner [70] can be adapted to approach system-wide reform of research practices (Table 6).

Without the connections and cooperation of the different levels identified in the social-ecological model [70], real reform improving research quality is unlikely to succeed. Barnett and Byrne [71] explain a bystander effect currently occurring in research quality, with everyone watching and waiting for someone else to act while the research systems further decline, with publishers expecting institutions to prevent and educate against poor practice, institutions expecting their staff to protect their reputations and for journals to improve the peer review, and funders willing to fund new research but not quality control. They recommend diverting 1% of publishers' profits and scientific funding to quality control [71]. Currently, no time or money is built into the system for research maintenance or quality control. As seen in preventative health [72], broad structural change is likely to occur only with investment and policy implementation.

**Table 6. Social Ecological Model a system-wide approach to reform statistical practices adapted from Bronfenbrenner [70].**

| Level | Description |
|---|---|
| Individual | - Individual characteristics that influence behaviour change, including knowledge, statistical literacy, beliefs, attitudes, and personal traits such as gender, years of experience and job security. |
| Interpersonal | - Formal and informal social networks and support systems that can influence individual behaviours to promote interpersonal growth that encourages good statistical practice including peers and co-workers, and mentors. |
| Community | - Formal or informal social norms can limit or encourage good statistical and research behaviours among individuals, groups, or organisations. |
| Organisational | - Organisations rules and regulations for operations, for example, ethics committees in hospitals or universities. |
| | - Access to research infrastructure, including statistical resources such as statistical programs and educational resources. |
| | - Access to qualified statisticians. |
| | - Sustainable research metrics with quality valued over quantity. |
| | - Organisational oversite of research misconduct. |
| Policy Environment | - Local, state, national and global policies regarding the allocation of resources for meta-research to tackle systematic issues. |
| | - Regulatory bodies, state, national and international integrity commissions, for the oversite of research misconduct. |

## Checklists and automated reviews

Our findings suggest that while most authors report regression coefficients, they often do not provide any measure of uncertainty around their result, and it can be challenging to identify the specific statistical method used. Journals can improve the quality of statistical reporting by implementing policies that standardise the presentation of statistical results. This could involve including all statistics in tables, whether in the main body of the paper or the supplementary materials, with clear identification of the statistical tests used.

Many journals require reporting guidelines, including statistical guidelines such as SAMPL [17]. This could be an opportunity for researchers to seek advice and improve statistical methodology. However, the current checklist approach of just providing page numbers instead of details has been criticised, with Blanco et al. [73] questioning whether checklists submitted by authors reflect the information presented in articles. They randomly selected 12 randomised controlled trials from three journals and found that only one article fully adhered to CONSORT guidelines. They concluded that journals needed action to ensure transparent reporting, including checking the items examined by editors or trained editorial assistants. *PLOS ONE* recommends that authors use SAMPL to provide guiding principles for reporting statistical methods and results and specific instructions for reporting linear regression; our results show that the guidelines are not widely followed. *PLOS ONE* also recommends the use of STROBE [32] for observational studies, but our results showed poor reporting of results with papers often lacking detail on whether the study was descriptive, associational, or predictive and a clear statement of what variables were selected and why. These results support Pouwels et al. [74], who concluded that authors should be required to submit the checklist with text excerpted from the manuscript instead of just referring to page numbers.

When journals introduce new policies, it's important to monitor their value. These policies should not just increase the author and reviewer burden without improving quality. There's a risk that researchers might provide normative responses to checklists rather than focussing on improving overall research quality [75]. This was evident in interventions promoting the better use of confidence intervals, where the impact on interpretation quality was minimal

[68,76]. To reduce this burden on reviewers, it's recommended that journals provide templates of papers with expected results and standard reporting. Reviewers can be provided with interactive checklists, similar to the one used in this research. For example, if reviewers indicate that confidence intervals were not reported, an automated feedback system can educate authors on table formatting and interpretation of results.

Some readers may ask, can statistical reviews be automated? While this is still a developing field, there have been previous attempts [77], including text mining and *statcheck*, an algorithm designed to scan papers to detect inconsistencies in calculated test statistics, degrees of freedom and their associated p-values [54]. Roughly half of the papers reviewed had at least one p-value that did not correspond with their associated test statistic and degrees of freedom. *Statcheck* can only process data with specific APA formatting (e.g. $R^2 = .84$, $F(1, 98) = 2.52$, $p < .001$), and was found to only process 61% of all statistical tests [78]; with Böschen [79] reporting that *Statcheck* should not be used to detect irregularities in statistical results as it is unable to deal with even small deviations in formatting. The current study found results were often incomplete and inconsistently formatted, reflecting differing reporting practices across health fields, a current barrier to automation.

Zhu et al. [80] recently tried to establish if Large Language Models (LLMs) are as good as statisticians. The authors used 11,623 examples to evaluate LLMs' proficiency in specialised statistical tasks, including identifying appropriate statistical methods and parameters. The authors found that postgraduate statistical students had an accuracy of 53% when performing statistical tasks, whereas GPT-4o outperformed other LLMs and humans with an accuracy of 65%. The authors found that LLMs are good at distinguishing different statistical tasks but may struggle to use domain knowledge, whereas humans are prone to task confusion; they recommended complementary use of LLMs with researchers [80].

Our study did not compare large language models to statistical reviewers. Instead, each paper received two statistical reviews which were compared to assess reliability. Statistical reviewers were highly reliable in identifying common statistics but less reliable for questions requiring interpretation. While not quantitatively measured, the primary author reviewed all papers and every disagreement and noted when there seemed to be an apparent reason for disagreement. These reasons can be broken into four areas: Concentration, Task confusion, Experience, and Paper Coherence.

LLMs are expected to outperform reviewers when errors are related to concentration, such as missing statistical information in the text, but have been found to have low accuracy when reviewing becomes more complex [81]. Some papers had very poor statistical reporting and were challenging to review, often leading statisticians to interpret their best guesses based on what the authors reported. Some methods sections were unclear and may not match what was reported in the results, and often had tables without in-depth interpretation or identification of statistical tests used. This may also be challenging for LLMs as statisticians use broad contextual knowledge developed over their careers to interpret statistical methods and can identify different mistakes commonly made depending on the subject area and statistical packages used. Future research should consider codifying this knowledge to enhance the performance of large language models. Therefore, while automated tools are helpful, further development is required to increase accuracy before being used to review papers [81], and should only be used to aid the reviewer, such as helping screen the paper for checklists. Reviewers can then use this information to improve the interpretation of results [77].

## Involvement of statisticians

The volume and statistical complexity of most medical research have increased drastically in the last couple of decades, with the use of advanced methods such as survival analysis and multivariable linear regression now commonplace [25]. Our study confirmed this, with most studies using multiple statistical methods and multivariable analysis. Unfortunately, serious flaws in published work are also commonplace, with Altman et al. [25] reporting that many of these problems are caused by statistical analyses performed by health professionals with an inadequate understanding of statistical methods. Studies in quality improvement consistently recommend that authors should involve biostatisticians in projects early, enabling well-designed studies with robust interpretation of results [64]. However, many projects either completely lack involvement from a biostatistician, or they are involved too late to improve study findings effectively. This has been a long-recognised phenomenon across all research fields, with Fisher [82] famously saying, "To consult the statistician after an experiment is finished is often merely to ask him to conduct a post-mortem examination. He can perhaps say what the experiment died of". This essentially highlights that no analytical methods can rescue the result once a study has been undertaken with poor design.

Our study highlighted that statistical sections were often generic, emphasising p-values rather than practical importance, with only (23%) of authors directly interpreting the size of regression coefficients, indicating statistical input may have improved reporting. The use of statistical expertise was examined by Altman et al. [64], who surveyed the authors of all original research articles submitted to the BMJ and Annals of Internal Medicine over five months in 2001. Authors were asked if studies had statistical or epidemiology input, the stage this occurred and reasons if none was used. Of the 704 authors who responded, 39% (273) of papers had input from a statistician. For the papers with statistical input, 30% of authors identified their first major contribution was at the analysis stage. Of these papers, a third of biostatisticians were not acknowledged for their work. Altman et al. [64] found that articles without methodological support were more likely to be desk-rejected (71%) than articles with statistical input (57%). A more recent survey by Sebo et al. [83] randomly selected 781 articles published in 2016 journals from high-impact medicine and primary care journals and found when a statistician is involved as a co-author, time to publication is reduced. Mullner et al. [84] reviewed 537 papers from medicine and found when statisticians are involved in studies, inadequate reporting of adjustment for confounders drops from 56% to 27%. While we did not measure the involvement of statisticians in our research, poor reporting was commonplace; statistical reviewers had difficulty identifying if there was a process for selecting variables or even whether there were linear regression results in the paper. Therefore, it is recommended that statisticians be involved early in medical research and be appropriately recognised for their contribution. While much of the burden of implementing checklists to improve statistical quality falls on journals, we recommend institutions such as universities take responsibility for their paper submissions, stopping the poor research before it gets to the journal. To achieve this goal, more funding from institutions for central statistical support is required. A report by the NHMRC (Australia's major funder of health and medical research) on research quality identified *study statistics and analysis as a critical core competency for high quality research*, and noted *statisticians as advisors to/members of ethics committees* as an example of *institutional support that will foster high quality research* [85]. There have also been broader calls for statisticians to be involved in all medical research, to improve study design and interpretation of results [86]. Involvement of statisticians should not be a tick box exercise, as it is crucial to engage statisticians with the appropriate expertise and contextual knowledge to meet the specific demands of research projects. For instance, developing

prognostic models or structural causal models requires specialised skills that not all statisticians possess. It is recommended, when possible, to use accredited statisticians, as professional bodies such as the Statistical Society of Australia bind members to a code of conduct that encourages them to offer services only within their professional competence and refrain from claiming expertise they do not possess. These guidelines support ethical practice and help researchers access high-quality statistical expertise, enhancing the reliability and rigour of research [87].

Post-publication peer review, as conducted by statisticians in the current study, allows for transparent and continuous research evaluation, identifying flaws or errors [88]. In an environment where digital technology is the norm, researchers can be given real-time feedback about statistical methods through journal websites, pre-prints and changes made through version control. There is an opportunity to change practice by encouraging researchers to take ownership of their errors, where publications are regarded as the beginning of the journey, and published work is viewed as dynamic 'living documents' that can be changed and updated as errors are identified [89]. For this to occur, both researchers and institutions need to invest in quality over volume, with negative perceptions about paper corrections overcome.

## Limitations

*PLOS ONE* is a large cross-discipline journal, but may not be representative of all health and biomedical journals. To obtain stable estimates given the low prevalence of some reporting behaviours, the sample was made in a single year rather than spanning multiple years. While this approach may limit generalisability, the study's findings align consistently with existing literature on researchers reporting statistical results. The focus of this study was linear regression, as there will be different misconceptions driving understanding in comparison to ANOVA. We focused on the interpretation of continuous variables, however, we recommend that future questionnaires be seen in a general linear model framework. Initially, the questionnaire contained 55 items and included an interpretation of categorical independent variables, but it was removed due to length concerns. However, breaking up the interpretation of coefficients into continuous and categorical, as well as adding if post hoc tests were used, would not compromise length but improve interpretability.

## Conclusions

Linear regression is one of the most frequently used statistical methods, so researchers should be able to interpret its output. Unfortunately, our research shows that the average researcher tends to over-rely on p-values and significance rather than the contextual importance and robustness of conclusions drawn. This systematic failure in statistical reporting highlights the need for investment in research training and quality control; this should be across the board, from ethics to submission of research to post-peer review, allowing qualified statisticians and other methodological experts to be involved through the entire project cycle. The research environment is an ecosystem, and future meta-research should consider how the different levels of the system interact, understand the behaviour of individuals, their support systems, the community, organisations, and policy environment, as well as adapt and use established knowledge about behaviour change. Journal policies can achieve improvements in basic reporting, with the recommendation from this study to introduce interactive checklists for authors and reviewers, so when poor reporting occurs, automated feedback can be provided with education on how the tables should be formatted and results interpreted. Journals could also produce template papers and standardise reporting for commonly used statistical tests. To increase the transparency of reporting, all statistical tests should be put into

tables, whether in the main body of the paper or the supplementary materials; it should be clear from tables what test was used and if models are univariate or multivariable. Finally, post-peer review needs to be encouraged, where correcting errors and clarifying research is rewarded rather than punished, and research papers are regarded as living documents with version control; for this to occur, there needs to be a cultural change and investment in how academics and institutions think about academic output, with research maintenance built into roles shifting away from volume to quality of research.

## Supporting information

**S1 Table.** Full reporting of agreement and reliability for statistical raters.
(DOCX)

## Acknowledgments

We acknowledge all the statisticians (named and not named) who kindly gave up their time to contribute to this publication by reviewing papers, including: Ingrid Aulike, Peter Baker, Brigid Betz-Stablein, Enrique Bustamante, Taya Collyer, Susanna Cramb, Alanah Cronin, Laura Delaney, Zoe Dettrick, Eralda Gjika Dhamo, Des FitzGerald, Peter Geelan-Small, Edward Gosden, Alison Griffin, Jenine Harris, Cameron Hurst, Kyle James, Helen Johnson, Jessica Kasza, Karen Lamb, Stacey Llewellyn, James Martin, Miranda Mortlock, Satomi Okano, Alan Rigby, Michael Steele, Megan Steele, Jacqueline Thompson, Simon Turner, Michael Waller, Kevin Wang, Jace Warren, Natasha Weaver, Lachlan Webb, and Janet Williams.

## Author contributions

**Conceptualization:** Lee Jones, Adrian Barnett, Dimitrios Vagenas.

**Data curation:** Lee Jones.

**Formal analysis:** Lee Jones.

**Funding acquisition:** Lee Jones.

**Investigation:** Lee Jones, Adrian Barnett, Dimitrios Vagenas.

**Methodology:** Lee Jones, Adrian Barnett, Dimitrios Vagenas.

**Project administration:** Lee Jones, Dimitrios Vagenas.

**Resources:** Lee Jones, Dimitrios Vagenas.

**Software:** Lee Jones.

**Supervision:** Lee Jones, Adrian Barnett, Dimitrios Vagenas.

**Validation:** Adrian Barnett, Dimitrios Vagenas.

**Visualization:** Lee Jones.

**Writing – original draft:** Lee Jones.

**Writing – review & editing:** Lee Jones, Adrian Barnett, Dimitrios Vagenas.

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
