## [Decision Letter · Decision Letter 0]

21 Nov 2024

PONE-D-24-21233Linear regression reporting practices for health researchers, a cross-sectional meta-research studyPLOS ONE

Dear Dr.  Jones,

Thank you for submitting your manuscript to PLOS ONE. After careful consideration, we feel that it has merit but does not fully meet PLOS ONE’s publication criteria as it currently stands. Therefore, we invite you to submit a revised version of the manuscript that addresses the points raised during the review process.

Your manuscript has been reviewed and requires modifications prior to making a decision. The comments of the reviewers are included at the bottom of this letter. The manuscript presents a well-conducted study that underscores the importance of statistical methods and transparent research reporting. While the study is valuable, some improvements are recommended: the theoretical aspects of regression could be shortened, as they are widely available in standard texts, and the role of statistics in health systems could be elaborated to provide better context. Additionally, the authors should clarify certain methodological choices, such as the selection of specific timeframes, and consider emphasizing the need for involving statisticians with relevant expertise throughout research projects to enhance rigor and reliability. These adjustments will further refine the manuscript and improve its clarity for readers.

We look forward to receiving your revised manuscript.

Kind regards,

Asli Suner Karakulah, PhD

Academic Editor

PLOS ONE

Journal Requirements:

 "There was no cost associated with this research except for attending conferences. These costs were covered by the primary author’s PhD allocation from the health faculty, Queensland University of Technology, and scholarships. The Statistical Society of Australia (SSA) and the Association for Interdisciplinary Meta-research & Open Science (AIMOS) supported the primary author with travel grants to attend their respective conferences. These scholarships did not influence the results of the study."

Reviewers' comments:

Reviewer's Responses to Questions

**Comments to the Author**

1. Is the manuscript technically sound, and do the data support the conclusions?

Reviewer #1: Yes

Reviewer #2: Yes

Reviewer #3: Yes

Reviewer #4: Yes

2. Has the statistical analysis been performed appropriately and rigorously? 

Reviewer #1: Yes

Reviewer #2: Yes

Reviewer #3: Yes

Reviewer #4: Yes

3. Have the authors made all data underlying the findings in their manuscript fully available?

Reviewer #1: Yes

Reviewer #2: Yes

Reviewer #3: Yes

Reviewer #4: Yes

4. Is the manuscript presented in an intelligible fashion and written in standard English?

Reviewer #1: Yes

Reviewer #2: No

Reviewer #3: Yes

Reviewer #4: Yes

5. Review Comments to the Author

Reviewer #1: A very well drafted manuscript by the authors. In the article they have clearly explained how the statistical methods were poorly reported without sufficient details to reproduce them and also how to improve the statistical quality and direct health funding to effective treatments.

Reviewer #2: It is an interesting study and underlines the importance of statistical test and the method of reporting the test and its output. However there are concerns which need to be addressed.

1. The authors can shorten the theoretical aspects of regression which can be obtained in standard statistical books.

Reviewer #3: Excellent piece of work. Just a couple of minor comments.

1. You only know what is *reported*. You say "Authors rarely check multicollinearity...". Probably true, but I know I (almost) always check multicollinearity but I don't think I have ever mentioned it in a publication.

2. PLOS ONE has no word limits, so authors should be able to report everything needed, but other medical journals often have very strict word limits and some still do not allow supplementary material. People learn to leave things out to fit word limits, and I htink this should be mentioned. It is part of the system-wide problem.

Reviewer #4: As a strong supporter of greater transparency and improved quality in research reporting, I welcome and appreciate meta-research studies that emphasise the need for higher research quality and better reporting.

I believe this study is a valuable contribution to this area, and I commend the authors for their appropriately designed and conducted work. The study methods are quite straightforward, and I don’t have any major remarks. I have only a few comments, which I hope will help make the article clearer for readers and suggest a few points the authors may want to consider.

INTRODUCTION

[Comment #1]

In the introduction (page 2), the author begins the first paragraph with: “Health systems are generally complex and often comprise a network of interrelated variables. Statistical methods can help untangle and understand these relationships, allowing the quantification and estimation of the effects of diseases and treatments.” This opening sentence is somewhat unclear. Health systems utilise results from studies on the effects of diseases and treatment effectiveness to inform decision-making. However, these decisions are also influenced by factors other than statistics (e.g., see the GRADE Evidence-to-Decision framework). While the role of statistics is unquestionably important, it is not the only factor. I suggest elaborating this a bit more, clarifying the role of statistics in health system decisions and policy, or changing the starting sentence. As presented, it seems that regression models are directly linked to uncovering the complexity of the healthcare system, whereas in the context of this study, they mainly refer to the statistical results of studies.

Moberg, J., Oxman, A.D., Rosenbaum, S. et al. The GRADE Evidence to Decision (EtD) framework for health system and public health decisions. Health Res Policy Sys 16, 45 (2018). https://doi.org/10.1186/s12961-018-0320-2

[Comment #2]

It may be worth mentioning that the quality of reporting is also important for assessing the risk of bias. Without sufficient information, the assessibility of the risk of bias is reduced.

Tikka C, Verbeek J, Ijaz S, et al. Quality of reporting and risk of bias: a review of randomised trials in occupational health. Occupational and Environmental Medicine 2021;78:691-696.

METHODS

- Sampling (randomisation)

[Comment #3]

It is unclear why 2019 was selected for the search. Could the authors elaborate a bit more on this decision? This choice can have implication of the generalisability of the findings. This should be probably also addressed in the discussion/limitation.

[Comment #4]

I cancelled some comments and requests for clarification after realising they were addressed in the medRxiv preprint. It seems this paper presents a shorter version of the method section from the preprint. I suggest mentioning this (e.g., that more details about the methods are available in the preprint) at the beginning of the methods section. However, I wonder why these details cannot be included in the current paper, as it might be more convenient for readers. Just a personal opinion.

- Comparison of results to other research

[Comment #4]

The paper by Strasak et al., referred to articles published before SAMPL was available. This could be mentioned and considered.

- Multicollinearity

[Comment #5]

The study did not differentiate whether the regressions were used for prediction/forecasting or causal inference. Multicollinearity is considered more problematic for (causal) inference than prediction. This may be discussed.

RESULTS AND DISCUSSION

- System-wide reform of statistical practices

[Comment #6]

This is more of a comment and not necessarily something the authors need to include in their article. However, I believe it should be emphasised more that the academic environment provides little incentive for high-quality research, where involving a statistician would be a consequential/logical step for conducting more rigorous research. Academia continues to prioritise quantity over quality for various reasons. While efforts such as sensitisation, the Social Ecological Model, or other initiatives like the Hong Kong Principles or CoARA are commendable, they do not appear to be effective. Only structural changes and enforced measures are likely to improve the situation. Maybe reputable national institutions such as the Statistical Society of Australia can press academic and governmental institution to implement (i.e., making mandatory) higher standards.

- Checklists and automated reviews

[Comment 7]

I completely agree with what is presented by the authors, including the suggestion that journals should require reporting the full text, not just the page, in quality of reporting checklists. The authors also mention text-mining-based programs (statcheck), which unfortunately do not work well. However, the use of LLMs could be useful and well-suited for this purpose. The authors may want to emphasise that LLMs could facilitate the development of automatic tools in the future.

This has also been suggested by Böschen, who criticised statcheck: https://arxiv.org/abs/2408.07948

https://doi.org/10.48550/arXiv.2408.07948

- Involvement of statisticians

[Comment 8]

I agree with the authors' main arguments and their encouragement to involve statisticians in research projects. However, it is important to emphasise that not all statisticians possess the necessary expertise for every project (i.e., it is not enough to involve a statistician to tick the box). Specific areas of expertise should be considered when involving experts. For example, as mentioned earlier, developing prognostic models or structural causal models requires statisticians with different areas of expertise and contextual knowledge. As an editor, I have observed cases where involving statisticians did not generate better results because the statisticians lacked relevant specific knowledge. While the authors may find this point obvious, I strongly believe it is not generally recognised. Highlighting this could be valuable.

- Conclusions

[Comment #9]

Related to my previous comments, I would modify “allowing statisticians and other methodological experts to be involved through the entire project cycle” to something like: “allowing QUALIFIED statisticians and other methodological experts in the specific area of investigation to be involved throughout the entire project cycle”

6. PLOS authors have the option to publish the peer review history of their article (what does this mean?). If published, this will include your full peer review and any attached files.

Reviewer #1: **Yes: **L Naga Rajeev

Reviewer #2: **Yes: **Dr Seema Patrikar

Reviewer #3: No

Reviewer #4: No

---

## [Author Response · Author response to Decision Letter 1]

13 Dec 2024

I have addressed all the reviewers comments, see the response to reviewers.

---

## [Decision Letter · Decision Letter 1]

29 Jan 2025

Linear regression reporting practices for health researchers, a cross-sectional meta-research study

PONE-D-24-21233R1

Dear Dr. Jones

We’re pleased to inform you that your manuscript has been judged scientifically suitable for publication and will be formally accepted for publication once it meets all outstanding technical requirements.

Kind regards,

Asli Suner Karakulah, PhD

Academic Editor

PLOS ONE

Additional Editor Comments (optional):

The authors addressed the reviewers' concerns and substantially improved the content of the manuscript. So, based on my own assessment as an academic editor, the manuscript can be accepted in its current form.

Reviewers' comments:

Reviewer's Responses to Questions

**Comments to the Author**

1. If the authors have adequately addressed your comments raised in a previous round of review and you feel that this manuscript is now acceptable for publication, you may indicate that here to bypass the “Comments to the Author” section, enter your conflict of interest statement in the “Confidential to Editor” section, and submit your "Accept" recommendation.

Reviewer #1: (No Response)

Reviewer #3: All comments have been addressed

Reviewer #4: All comments have been addressed

2. Is the manuscript technically sound, and do the data support the conclusions?

Reviewer #1: (No Response)

Reviewer #3: Yes

Reviewer #4: Yes

3. Has the statistical analysis been performed appropriately and rigorously? 

Reviewer #1: (No Response)

Reviewer #3: Yes

Reviewer #4: Yes

4. Have the authors made all data underlying the findings in their manuscript fully available?

Reviewer #1: (No Response)

Reviewer #3: Yes

Reviewer #4: Yes

5. Is the manuscript presented in an intelligible fashion and written in standard English?

Reviewer #1: (No Response)

Reviewer #3: Yes

Reviewer #4: Yes

6. Review Comments to the Author

Reviewer #1: (No Response)

Reviewer #3: This was a good paper before revision, and is now better. I thought the instructions above said i could skip the comments to author, but it seems impossiblet to submit the review without writing something here.

Reviewer #4: I thank the authors for thoroughly addressing my comments. I appreciate their important contribution to improving reporting and methodology in scientific publications.

7. PLOS authors have the option to publish the peer review history of their article (what does this mean?). If published, this will include your full peer review and any attached files.

Reviewer #1: **Yes: **Dr L Naga Rajeev

Reviewer #3: No

Reviewer #4: No

---

## [Editor Report · Acceptance letter]

PONE-D-24-21233R1

PLOS ONE

Dear Dr. Jones,

I'm pleased to inform you that your manuscript has been deemed suitable for publication in PLOS ONE. Congratulations! Your manuscript is now being handed over to our production team.

Kind regards,

on behalf of

Dr. Asli Suner Karakulah

Academic Editor

PLOS ONE